# C- and L-Bands Wavelength-Tunable Mode-Locked Fiber Laser

**Jiajing Lang [1,2], Cheng Chen [1,2], Pu Zhang [3], Mei Qi [4,*] and Haowei Chen [1,2,*]**

1   Statel Key Laboratory of Energy Photon-Technology in Western China, International Collaborative Center on Photoelectric Technology and Nano Functional Materials, Institute of Photonics & Photon-Technology, Northwest University, Xi'an 710127, China; 202220844@stumail.nwu.edu.cn (J.L.); chencheng0@stumail.nwu.edu.cn (C.C.)
2   Shaanxi Engineering Technology Research Center for Solid State Lasers and Application, Shaanxi Provincial Key Laboratory of Photo-Electronic Technology, Northwest University, Xi'an 710127, China
3   State Key Laboratory of Transient Optics and Photonics, Xi'an Institute of Optics and Precision Mechanics, Chinese Academy of Sciences, Xi'an 710119, China; zhangpu@opt.ac.cn
4   School of Information Science and Technology, Northwest University, Xi'an 710127, China
*   Correspondence: qm@nwu.edu.cn (M.Q.); chenhaowei0320@163.com (H.C.)

**Abstract:** We report a single-wavelength tunable mode-locked fiber laser. The single wavelength can be tuned from 1537.49 nm to 1608.06 nm by introducing a Sagnac loop filter. As far as we know, this is the widest single-wavelength tuning range achieved in an erbium-doped mode-locked all-fiber laser based on nonlinear amplifying loop mirror (NALM). The laser's pulse width changes from 549 fs to 808 fs throughout the tuning process, the maximum average output power is 5.72 mW, and the single-pulse energy is 0.34 nJ at a central wavelength of 1556.53 nm. This laser source can serve as an efficient tool for applications that require a broad tunability range. The combination of femtosecond pulses and extensive wavelength tuning capabilities makes this laser system highly valuable in fields such as fiber optic communications, spectroscopy, sensing, and other applications that benefit from ultrafast and tunable laser sources.

**Keywords:** mode-locked; wavelength tunable; C- and L-bands





## 1. Introduction

Since the introduction of fiber in 1966, significant efforts have been made to extend the wavelengths of ultrafast lasers to increase communication capacity and cater for a wider range of potential applications. The development of amplifying systems across multiple bandwidth ranges has also gone into overdrive [1–4]. The C-band (1530–1565 nm) is currently used in telecommunications systems but has limitations in terms of channel capacity. Therefore, there is a need to extend this optical transmission window to increase the transmission capacity. This need has fueled substantial interest in the potential applications of fiber lasers operating in the L-band (1565–1625 nm) for various fields, including fiber optic communications, sensing, spectroscopy, and biomedical applications [5,6]. Recently, tunable mode-locked fiber lasers have become one of the fundamental light sources in numerous fields, such as materials processing [7], fiber-optic sensors [8], optical signal processing [9], and optical communications [10]. The progress of the tunable mode-locked fiber lasers has opened up new possibilities for versatile and efficient light sources with wide wavelength coverage, which has contributed to advancements in various technological areas.

Passive mode-locking technology is currently the most widely used and studied technique for generating ultrashort optical pulses, due to its advantages. The main passive mode-locking techniques involve two types: the first is saturable absorbing materials, containing a semiconductor saturable absorber mirror (SESAM) [11,12], graphene [13], and carbon nanotubes [14]. The second is artificial saturable absorbers, including nonlinear polarization rotation [15], a nonlinear optical loop mirror (NOLM) [16], and a nonlinear

amplifying loop mirror (NALM) [17]. Artificial saturable absorbers are more appropriate for use in lasers to achieve tunable performance because of the inherent additional filtering effect [18,19]. The asymmetric amplification present in NALM, in contrast to NOLM, makes it extremely useful for optical frequency combs and high repetition frequency pulse generation.

As the tuning characteristics of a laser depend largely on the tunable filter, much effort has been put into building filters with excellent performance to expand the tunable fiber laser. So far, various filtering techniques have been explored, including Lyot filters [20,21], reflection gratings [22,23], fiber optic gratings [24], Sagnac loop filters [11,25–28], volume Bragg gratings [29], and intracavity birefringence filtering effects [30,31], etc. Nyushkov B. et al. achieved tuning of the output radiation wavelength from 1524 nm to 1602 nm over a range of 78 nm using a reflection diffraction grating [22]. Armas-Rivera. I et al. achieved tunable mode-locking in the range of 26.3 nm by temperature controlling the PMF from 1543.2 nm to 1569.5 nm using a Sagnac ring [11]. Wu et al. continuously tuned the wavelength from 1544.1 nm to 1560.8 nm in a ring mode-locked SESAM-based fiber laser using a birefringent Sagnac filter [27]. Tao et al. achieved a wavelength-tunable, pulse-type-switchable fiber laser with tuning wavelengths from 1548 nm to 1564 nm by introducing polarization-maintaining er-doped fiber (EDF) into a Sagnac ring and combining it with NALM [28].

Incorporating a tunable filter device into the laser cavity is a simple and reliable method for wavelength tuning. All-fiber tunable filters such as the Sagnac filter, which can achieve tunability while maintaining an all-fiber structure, are becoming a hot research topic because of their simple and compact structure, low cost, low loss, and high stability. However, there are few studies on single-wavelength continuous tunability of the C- and L-bands based on Sagnac filters.

In this paper, we report on a single-wavelength tunable mode-locked fiber laser based on NALM, which operates in both C- and L-bands. The single wavelength is tunable from 1537.49 nm up to 1608.06 nm in a range of 70.57 nm using the Sagnac filtering effect of the two-segment PMF, the widest single-wavelength tuning range ever achieved by an erbium-doped all-fiber laser based on the NALM. We believe that this mode-locked fiber laser with wide wavelength tunability and femtosecond pulse duration at C- and L-bands can be considered as an effective light source for a wide range of applications.

## 2. Experimental Setup and Principle

### 2.1. Experimental Setup

Figure 1 displays a wavelength tunable fiber laser schematic based on NALM mode-locking. The setup contains a unidirectional ring (UR) and a NALM ring, which is connected to the UR through a $2 \times 2$ optocoupler (OC) for a 40:60 coupling ratio. The NALM ring consists of a laser diode (LD) of a maximal output power up to 600 mW, a 980 nm/1550 nm wavelength division multiplexer (WDM), 0.6 m EDF (LIEKKI, ER110-4/125, Absorption at 1530 nm: $110.0 \pm 10.0$ dB/m), 8.2 m single-mode fiber (SMF, SMF-28e), two sections of 0.08 m PMF, and two polarization controllers (PCs). The PCs are used for regulating the light polarization states and the birefringence in the cavity for wavelength tuning. The UR consists of a PC, a 10:90 OC, as well as a polarization-independent isolator (PI-ISO), 10% of the power is being output, and 90% continues to circulate in the cavity. For the cavity, all other fibers are standard SMF, and the cavity has a total length of 12.2 m.

The pulse signals are measured with an ultrafast photodetector (Thorlabs DET08CFC, Newton, NJ, USA) that is linked to an oscilloscope (Agilent Technology, DSO9104A, Santa Clara, CA, USA). An optical spectrum analyzer (OSA, Yokogawa, AQ6370C, Tokyo, Japan) served to measure the pulse spectrum. A radio frequency (RF) spectrum analyzer (Keysight, N9000BCXA signal analyzer, Colorado Springs, CO, USA) and an autocorrelator (APE, PulseCheck-50, Berlin, Germany) served for measuring RF signals and pulse widths, respectively.

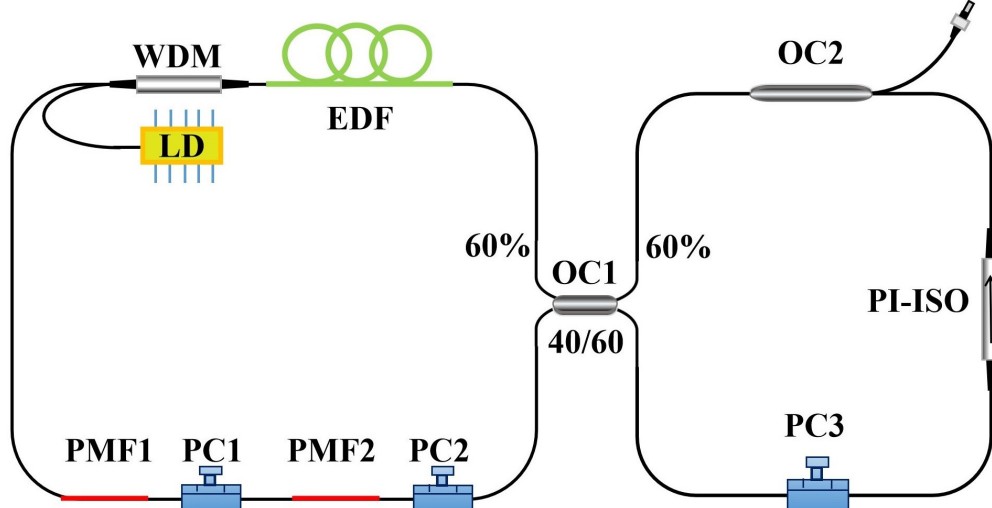

**Figure 1.** Schematic diagram of wavelength-tunable fiber laser based on NALM mode-locking. LD: laser diode; WDM: wavelength division multiplexer; EDF: er-doped fiber; PC: polarization controller; PMF: polarization-maintaining fiber; OC: optical coupler; PI-ISO: polarization-independent optical isolator.

*2.2. Experimental Principle*

In our experiments, the wavelength tunable is a filtering effect by adding two PMF segments into the Sagnac filter. The reason we use two sections of PMFs and two PCs is because a section of PMF and a PC can only increase or decrease stress birefringence individually. However, the transmission spectrum is affected by more factors when using two PMFs and PCs (see Equation (1)), so the transmission spectrum is more variable and simpler to achieve wavelength tuning. The transmission spectrum of the filter can be written as [32,33]:

$$T = (1 - 2k)^2 + 4k(1-k)\left( \begin{array}{l} \sin(\varphi_1 + \varphi_2)\sin\left(\frac{\theta_1+\theta_2}{2}\right)\cos\left(\frac{\theta_1-\theta_2}{2}\right) \\ + \sin(\varphi_1 - \varphi_2)\cos\left(\frac{\theta_1+\theta_2}{2}\right)\sin\left(\frac{\theta_1-\theta_2}{2}\right) \end{array} \right)^2 \qquad (1)$$

where $\theta_1$ and $\theta_2$ are the relative angles among the light polarization direction and the fast and slow axes of each PMF, which can be altered by rotating the PCs. $\varphi_n$ is the phase shift caused by the birefringence of PMF, and it can be expressed using the formula below: $\varphi_n = \pi \Delta n L_n / \lambda (n = 1, 2)$. $\lambda$ is the central wavelength; $L_1$ and $L_2$ are the lengths of PMF1 and PMF2, respectively; $\Delta n$ is the birefringence of the PMF, which is set to be $4.8 \times 10^4$. In the experiment, $L_1 = L_2 = 0.08$ m. Thus, the simulated transmission spectrum for the filter obtained is shown in Figure 2a, with a free spectral range (FSR) of about 31.3 nm. The adjacent curves represent the filter curves at different birefringence coefficients, and it is evident that wavelength tuning can be achieved by inducing only a weak birefringence change on the order of $1.5 \times 10^{-7}$. This illustrates that a small change in the birefringence can cause large variations in the laser's output wavelength, demonstrating excellent wavelength tuning capability. The measured typical transmission spectrum is shown in Figure 2b, with an FSR of about 31 nm. The different colours represent the variation of the filtered spectrum in different polarization states in the experiment. It can be seen that the FSR obtained from the simulation corresponds well to the experimental results. The amplified spontaneous emission (ASE) spectrum of the EDF is presented as an inset in Figure 2b.

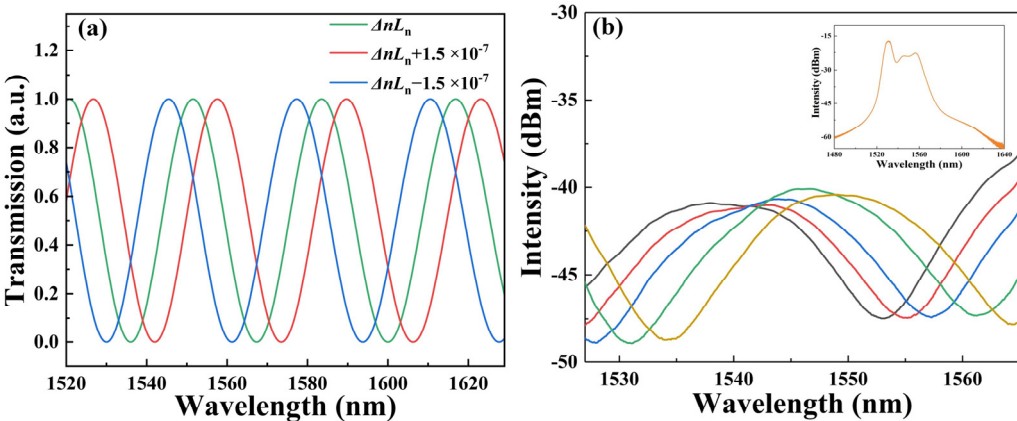

**Figure 2.** (**a**) Filtering curves under stress birefringence effects. (**b**) Experimentally measured filtering spectrum (inset: the amplified spontaneous emission of EDF).

## 3. Experimental Results and Discussion

Benefiting from NALM's excellent mode-locking performance, when reaching 200 mW of pump power, the PCs can be properly adjusted to achieve stable traditional soliton mode-locking. The typical mode-locked at 200 mW, as depicted in Figure 3. Figure 3a displays the corresponding pulse sequence with 59.5 ns pulse interval. The upper inset shows the oscilloscope trace in the 100 ms time domain, indicating the highly stable mode-locking. The spectrum in Figure 3b reveals the presence of symmetrical Kelly sidebands on each side on the spectrum, indicating that the laser works in a region of anomalous dispersion. The central wavelength is 1590.21 nm, and the corresponding 3 dB bandwidth is 4.02 nm. For the output pulse, the RF spectrum is presented, revealing a fundamental frequency is 16.81 MHz, with a signal-to-noise ratio of 69.46 dB, consistent with the length of the cavity, as shown in Figure 3c. The inset shows an RF spectrum spanning 0 to 500 MHz, further demonstrating mode-locked state stability. Immediately afterward, we measure the pulse's autocorrelation trace, as illustrated in Figure 3d. The measured autocorrelation trace has a pulse width of 673 fs under Sech$^2$ fitting, with the time-bandwidth product (TBP) of 0.32, which indicates a minor amount of chirp within the pulse.

Figure 4 shows the tunable mode-locked single-wavelength, and the corresponding autocorrelation traces for a pump power of 200 mW. Stable single-wavelength mode-locking is achieved through coarsening of the PCs. Fine adjustments of PC1 and PC2 achieve single-wavelength tuning operation. Therefore, the central wavelength of the mode-locked spectrum by continuously adjusting the PCs can be adjusted from 1537.49 nm to 1608.06 nm with a tuning range up to 70.57 nm, as illustrated in Figure 4a,c. Throughout the tuning process, the spectral profile undergoes a slight change, and the 3 dB bandwidth varies in the range 4.11 nm to 5.32 nm. It is worth noting that when the central wavelength is 1608.06 nm, a continuous wave (CW) can be seen near 1537 nm and 1564 nm, which is the result of a combined effects of loss, gain, and filtering [34] in the cavity. Firstly, the low-loss cavity can support L-band laser emission [35]. When achieving mode-locking at the central wavelength of 1608.06 nm, intracavity loss is at a lower loss compared to the C-band. Secondly, due to the strong gain competition in 1530 nm–1560 nm (The inset in Figure 2b) and the relatively high transmission rate in this bandwidth (see Figure 2), the presence of CW coexistence occurs near the wavelengths of 1537 nm and 1564 nm [36]. Figure 4b,d displays the measured autocorrelation traces corresponding to the respective central wavelengths. Observing the results, it is evident that as the central wavelength shifts from shorter to longer wavelengths, there is an accompanying increase in the pulse width, ranging from 549 fs to 808 fs. This change in pulse width can be attributed to variations in the intracavity polarization state and loss.

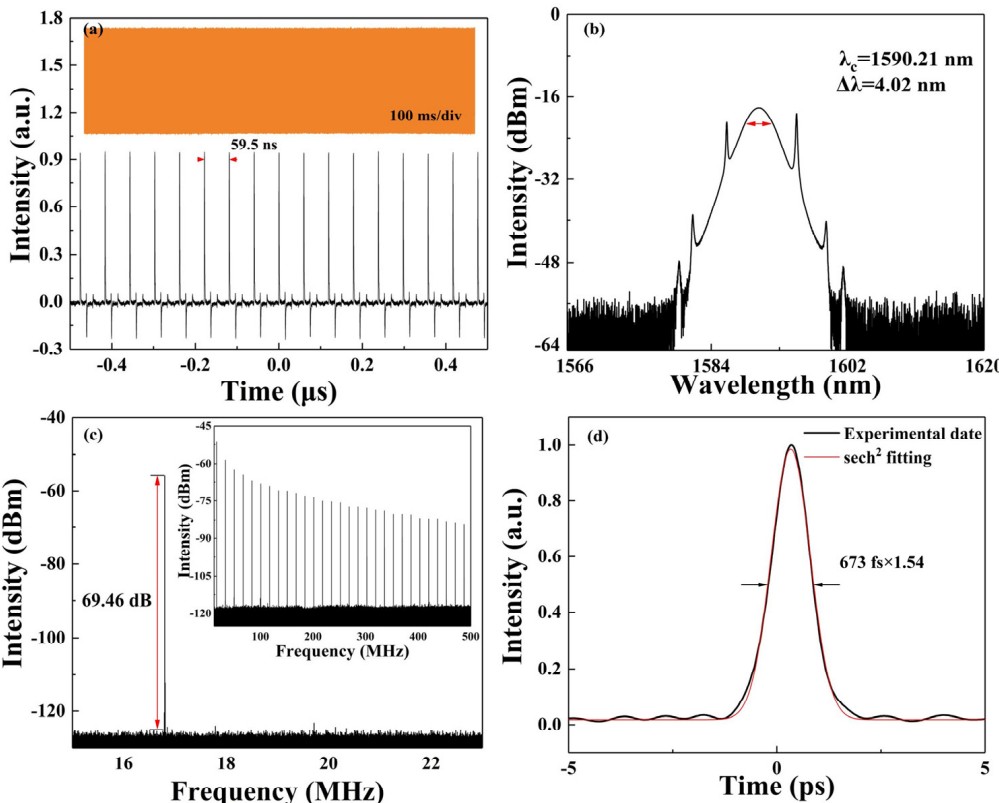

**Figure 3.** Typical mode-locked characteristics at 200 mW pump power. (**a**) Oscilloscope pulse traces; (**b**) Spectrum of the output pulses; (**c**) RF spectrum (inset: broadband RF spectrum in 500 MHz range); (**d**) Corresponding autocorrelation trace.

Many factors contribute to this phenomenon. On the one hand, different wavelengths have different polarization states within the cavity, and the transmittance magnitude of the components affects the overall loss in the cavity, which means that the peak intensity and the bandwidth of the artificially saturable absorber transmission curve vary with different polarization states; on the other hand, different operating wavelengths lead to differences in the gain intensity and dispersion of the laser cavity. All of them influence the pulse evolution and ultimately lead to different pulse characteristics. By adjusting the PCs in the experiment, we can control the laser's operating state and output wavelength, which can be tuned to the L-band. The ability to make a fiber laser operate in the L-band is firstly determined by the gain distribution within the cavity. This is followed by the operating conditions, for example, the pump power plays an important role because, as the operating wavelength deviates further from the maximum gain wavelength, higher pump power is required for energy compensation. Thus, the tunability of the central wavelength that can be achieved in the C- and L-bands is determined by the combination of the transmission spectrum of the filter and the ASE distribution of the EDF. It is evident from the ASE spectrum of the EDF in the inset of Figure 2b that there is a gain near the 1600 nm wavelength despite the small amplitude. By manipulating the PCs, we can control the laser's operating state and achieve tunability of the central wavelength in the C- and L-bands' spectral range.

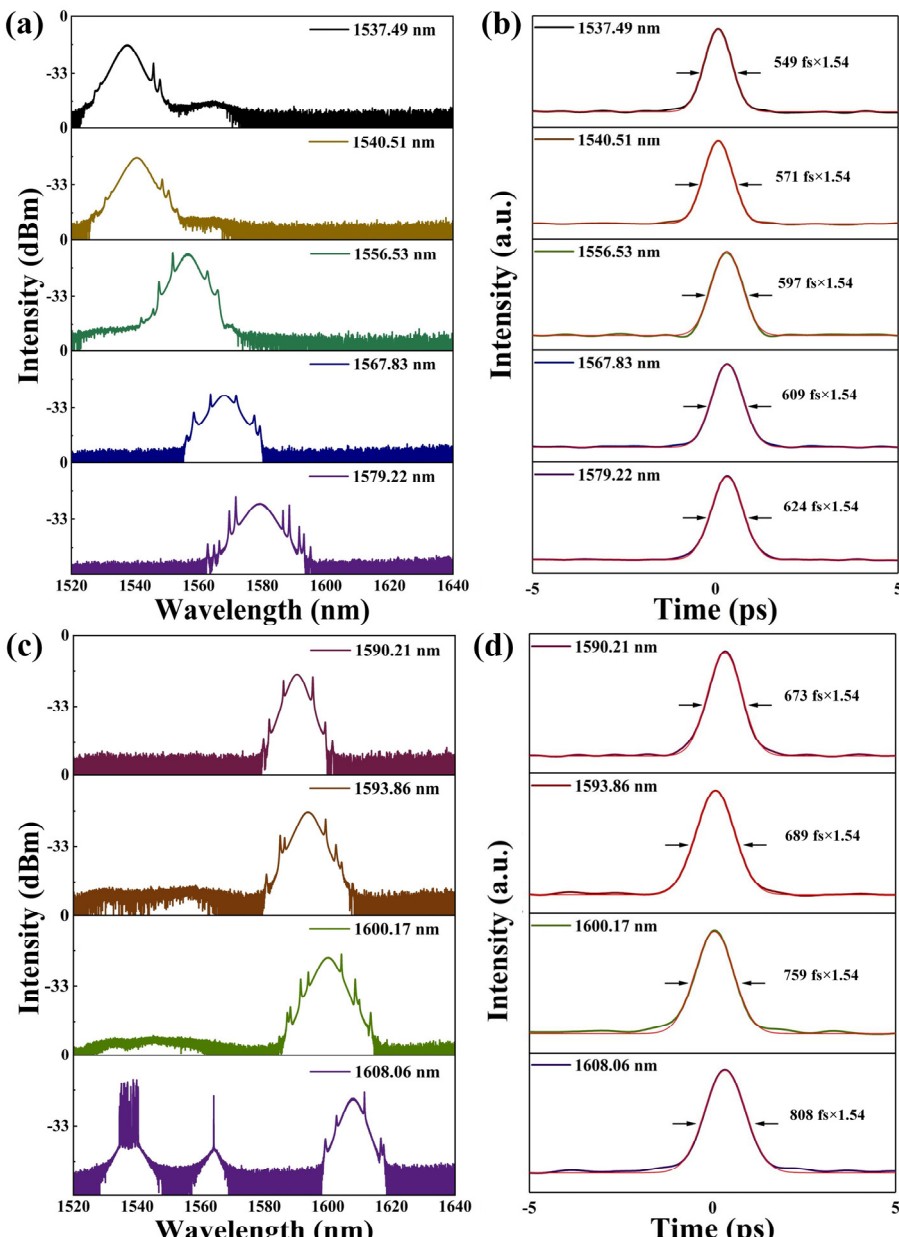

**Figure 4.** On the left, (**a**,**c**) are spectrum with tunable wavelength; the right columns (**b**,**d**) are the corresponding pulse widths collected by the autocorrelator.

To gain a much understanding of the output capacity and stability of the pulses at different operating wavelengths, we measure the output power, the TBP of the corresponding wavelengths, and the spectral stability, as shown in Figure 5. In Figure 5a, it can be observed that, as the central wavelength changes up to 1608.06 nm from 1537.49 nm, the average output power increases from 3.92 mW to 5.72 mW and then decreases gradually. At the wavelength of 1608.06 nm, the average output power decreases to 3.84 mW. The maximum average output power corresponds to the central wavelength of 1556.53 nm. There are two main reasons for this variation in output power. The first reason is that the gain distribution of EDF peaks around 1550 nm, with low gain at more distant wavelengths. As the central wavelength moves away from 1550 nm, the gain decreases, resulting in a reduction in the output power. The second reason is related to the intracavity fiber device, which has minimal loss at 1550 nm and a small operating bandwidth at low loss. Therefore, it has a low transmission loss at around 1550 nm, and the corresponding transmission loss increases as the central wavelength deviates further from 1550 nm. This increased loss also

contributes to the decrease in output power at 1608.06 nm. Figure 5a also presents the TBP at different wavelengths, ranging from 0.33 to 0.47, indicating that the solitons exhibit little chirping. Furthermore, the stability of the spectrum is demonstrated in Figure 5b, where we record the spectral fluctuations at 1603.72 nm over 1 h with a wavelength drift of only 0.0267 nm. This indicates excellent spectral stability of the laser output. Overall, the laser has high stability, wide tunability, and the widest tuning range for a single wavelength achieved in an all-fiber laser based on NALM mode-locking.

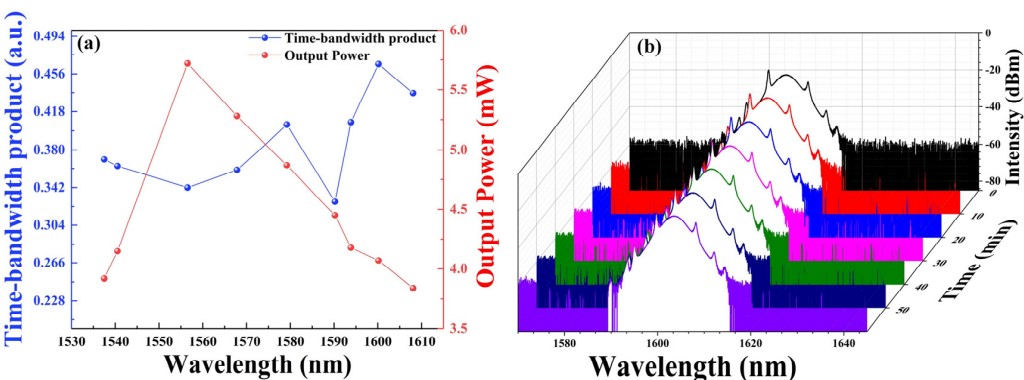

**Figure 5.** (**a**) Time-bandwidth product and output power versus wavelength; (**b**) Spectral stability.

## 4. Conclusions

In summary, a single-wavelength tunable mode-locked fiber laser based on NALM is reported. Due to the filtering characteristics of the Sagnac, simple adjustments allow for single wavelength tuning. The pulse width during single-wavelength tuning is also constantly changing. This is the widest single-wavelength tuning of any erbium-doped all-fiber laser based on NALM mode-locking, ranging up to 70.57 nm and covering both the C- and L-bands. We believe that this mode-locked fiber laser with femtosecond pulse duration and wide wavelength tunability at C- and L-bands can be considered as an effective light source for a wide range of application requirements.

**Author Contributions:** All authors listed have made a substantial, direct, and intellectual contribution to the work and approved it for publication. All authors have read and agreed to the published version of the manuscript.

**Funding:** Shaanxi Key Science and Technology Innovation Team Project (2023-CX-TD-06); Natural Science Foundation of China (62075237).

**Institutional Review Board Statement:** Not applicable.

**Informed Consent Statement:** Not applicable.

**Data Availability Statement:** The raw data supporting the conclusions of this article will be made available by the authors, without undue reservation.

**Acknowledgments:** The authors would like to thank the editors and reviewers for their efforts in supporting the publication of this paper.

**Conflicts of Interest:** The authors declare no conflict of interest.

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
