# Peer review of "C- and L-Bands Wavelength-Tunable Mode-Locked Fiber Laser"

_photonics, doi:10.3390/photonics10121379_

Round 1

Reviewer 1 Report

Comments and Suggestions for Authors

In the article the authors report about erbium-doped mode-locked fiber laser that can be tuned in a wide range from 1537.49 nm to 1608.06 nm. It based on nonlinear amplifying loop mirror and birefringent spectral filter. The article is quite interesting as a really wide tuning range has been demonstrated, however it has some drawbacks that should be addressed before publication.

1. In the abstract: achieved pulse energy should also be reported.

2. In the introduction when the authors addressed to using reflection gratings as a filter they refer only to ytterbium-doped lasers (refs [13] and [14], line 41). I believe the erbium-doped one may be more relevant. See, e.g. [1*] where a strong dependency of the generation regime from the filter contrast has been found.

[1*] I. S. Zhdanov, A. E. Bednyakova, V. M. Volosi, and D. S. Kharenko, “Energy scaling of an erbium-doped mode-locked fiber laser oscillator,” OSA Contin., vol. 4, no. 10, p. 2663, 2021.

3. The parameters (absorption, gain, core diameter) or/and the type of the EDF fiber should be listed in the experimental setup description section (line 89).

4. Fig.2. The transmission function should be presented in a whole range (up to 1620 nm).

5. Line 120. The authors use two identical PM fiber inside the cavity, but they produce exactly the same FSR of 31.3 nm and should not support lasing around 1600 nm due to strong amplification near 1530 nm. Please describe the reasons of such lengths in details.

6. Fig.4c should be presented in the same range as Fig.4a to eliminate the noise floor near 1530 nm. Looks like some background must definitely exist.

Reviewer 2 Report

Comments and Suggestions for Authors

Mode-locked fiber laser continuously tunable in a wide spectral range are of interest for many applications. This is why this matter had been intensively and extensively explored for decades, resulting in hundreds already published achievements. Therefore, it is wondering if it is still possible to develop a novel approach or innovation in this field.

Despite the fact that the technical content of the submitted manuscript is rather solid, and its quality of presentation is high enough, I did not find clearly stated scientific novelty, technological innovation, or another serious advantage in the presented laser and its characteristics compared to the earlier developments. In my opinion, to make this manuscript definitely acceptable for publishing in Photonics, it will be right if the author emphasizes in a more clear and evident way scientific novelty, technological innovation, or great advance in performance compared with the earlier published wavelength-tunable mode-locked fiber lasers.

For instance, a passively mode-locked Er-fiber laser reported in [ https://doi.org/10.1109/JLT.2019.2893291 ] provided femtosecond output continuously tunable even in a noticeably wider spectral range (from 1524 to 1602 nm). This work was not even cited in the present manuscript.

Reviewer 3 Report

Comments and Suggestions for Authors

 This manuscript reported a single-wavelength tunable mode-locked fiber laser. By introducing a Sagnac 11 loop filter, the single wavelength can be tuned from 1537.49 nm to 1608.06 nm. It can be considered to be accepted after minor revision.

1.Why two sections of 0.08 m PMF were used? What is the fusion angle between these two PMF?

2.The author wrote: A time-bandwidth product (TBP) of 0.32 is obtained, which indicates that a minor amount of chirp within the pulse. Why the output pulses have minor chirp?

Comments on the Quality of English Language

Minor editing of English language required

Reviewer 4 Report

Comments and Suggestions for Authors

The paper entitled “C- and L-bands wavelength-tunable mode-locked fiber laser” which reports a NALM-based mode-locked fiber laser with wide tunability. By introducing a Sagnac loop filter with two PMFs, the laser can be tuned from 1537.49 nm to 1608.06 nm with a single wavelength up to 70.57 nm. This continuously tunable laser in the C- and L-bands can be used as an effective source of light for many applications such as fiber-optic communication, sensing, spectroscopy and so on. This paper can be accepted, but some modifications are needed and related suggestions are given below:

1.      Is the free spectral range in Fig. 2(a) changed by 1.5×10-7 for stress birefringence for both PMF segments? If it is the same, which cannot be controlled to make it the same in the experiment, how does the filtered spectrum change when the stress birefringence applied to the two PMF segments is not the same?

2.      Why is there a small fraction of transmitted light in the first two spectrums compared to the other spectrums during the wavelength tuning process in Figure 4(a)?

3.      Why is an asymmetry present in the Kelly sidebands of the tuning process?

4.      The third paragraph of the experimental results section, “The transmittance magnitude of the components affects the overall loss in the cavity.” and “The ability makes a fiber laser to operate in the L-band is ……within the cavity” are not well connected, and it is suggested to modify them.

5.      Some formatting problems

(1) Line 132, “Fig. 2 (b)” has multiple spaces in the middle and needs to be checked.

(2) The font size of the word “autocorrelator” at the end of the Figure 4 note is not consistent with the previous one.

Comments on the Quality of English Language

Please further polish the English.

Round 2

Reviewer 1 Report

Comments and Suggestions for Authors

Almost all my comments were taken into account except the 5-th one, about two identical PM fiber inside the cavity. The authors explain the reason very well, but have not changed the article. I believe that a small explanation should be also added into the text to avoid possible misunderstanding among future readers.
